# Analysis of Running in Wilson’s Disease

**DOI:** 10.3390/sports10010011

**Published:** 2022-01-07

**Authors:** Sara Samadzadeh, Harald Hefter, Osman Tezayak, Dietmar Rosenthal

**Affiliations:** 1Department of Neurology, University of Düsseldorf, 40225 Düsseldorf, Germany; sara.samadzadeh@yahoo.com (S.S.); tezayak@hispeed.ch (O.T.); dietmar.rosenthal@med.uni-duesseldorf.de (D.R.); 2Department of Psychiatry, Psychiatriezentrum Kreuzlingen, 8280 Kreuzlingen, Switzerland

**Keywords:** Wilson’s disease, running, ground reaction forces, central pattern generator, severity of symptoms, optimization of therapy

## Abstract

Aim of the study was to analyze the ability of long-term treated patients with Wilson’s disease (WD) to run a distance of 40 m. 30 WD-patients from a single center were consecutively recruited. All patients were able to walk a distance of 40 m without walking aids. Vertical ground reaction forces (GRF-curves) were analyzed by means of an Infotronic^®^ gait analysis system (CDG^®^) and correlated with clinical and laboratory findings. Results of the WD-patients were compared to those of an age-and sex-matched control group. 25 of the 30 WD-patients were able to run. Patients being unable to run had a significantly (*p* < 0.03) higher non-motor score. In comparison to the controls speed of running was significantly (*p* < 0.02) reduced in WD-patients. Their duration of foot contact on the ground lasted significantly (*p* < 0.05) longer. Running was more irregular in WD and the variability of times to peak of the GRF-curves was significantly (*p* < 0.05) increased. All running parameters extracted from the GRF-curves of the CDG^®^ did not correlate with severity of WD. Cadence of running was significantly (*p* < 0.03) negatively correlated with serum liver enzyme levels. Running appears to be rather unimpaired in long-term treated WD, only 16% of the 30 WD-patients were unable to run. This knowledge is highly relevant for the patient management, but because of the missing correlation with severity of WD, analysis of running is of minor importance for monitoring WD-therapy.

## 1. Introduction

Wilson’s disease (WD) is a rare, recessively inherited deficit of copper transportation [1,2,3] leading to increased serum levels of free copper and progressive copper intoxication of multiple organs [4]. In the central nervous system (CNS) the basal ganglia, the cerebellum as well as brainstem nuclei are mainly disturbed [5]. A broad spectrum of neurological symptoms as parkinsonian gait, tremor, dysarthria, dystonia, and chorea results [6]. Relative frequencies of initial neurological symptoms have been reported in several large case series [7,8,9] and vary widely [6]. The reported rates for abnormal gait e.g., vary between 10 [9] and 75% [10,11].

Neurological symptoms and brain dysfunction respond to copper elimination therapy [12,13]. In a study on the prognostic significance of neurologic examination findings in Wilson disease [6] initial gait abnormalities turned out to predict the change of the used total neurological score the best. However, there was no correlation between initial and final gait scores, indicating variable response of gait abnormalities to therapy. In a recent study on the spectrum of neurological symptoms in long-term treated WD-patients [14] only 20% of the patients had a clinically manifest abnormality of gait.

Gait is a complex bipedal motor task which is under the control of several CNS structures, especially the basal ganglia, the cerebellum and the brainstem nuclei [15,16,17]. Therefore, gait disturbances have to be expected in neurological WD. Quantitative measurement of free walking confirms persistent gait abnormalities in long-term treated neurological WD [18]. These WD-patients who were diagnosed at the age of 22 in the mean walked slower than age-and sex-matched controls, but the temporal pattern controlled by the spinal and brainstem central pattern generators of walking turned out to be unimpaired [15,18]. In another study on gait disturbances in WD frequent falls were reported in a series of about 100 WD-patients [19]. In contrast to the mildly affected WD-patients mentioned before these patients were diagnosed at the age of 36 in the mean. This underlines the impact of early diagnosis and early onset of therapy on outcome, especially on gait in WD.

Running is an even more complex bipedal motor task than free walking [20]. In contrast to walking, the center of gravity is only intermittently supported by one leg during running. These stance phases are separated by intervals during which both legs are in the air. This implies that the musculature of one leg and especially the forefoot has to manage the complex motor task to take over the entire body weight without any help of the other side and to push immediately back to the other side thereafter. Our hypothesis, therefore, is that running is frequently disturbed in WD even in long-term treated patients with mild gait impairment.

So far little is known about the ability of patients with long-term treated WD to perform sports activities. However, this is an important aspect of quality of life for these young patients who are severely affected when they are diagnosed.

In general, clinical studies on running of neurological patients are rare [20,21]. During clinical neurological examination, running is usually not tested. Therefore, it does not surprise that no study on running in Wilson´s disease is available so far.

## 2. Materials and Methods

The study was performed according to the Declaration of Helsinki and the guidelines for good clinical practice (GCP) and approved by the local ethics committee of the University of Düsseldorf (Düsseldorf, Germany; study number: 5171).

### 2.1. Patients and Controls

All WD-patients who were continuously treated at Wilson’s disease out-patient department of the Neurological Clinic of the University of Düsseldorf were informed on the purpose of this study. Inclusion criteria were: (i) age > 18, (ii) patient not under care, (iii) diagnosis of WD confirmed in our center, (iv) regular treatment of WD for at least 2 years, (v) ability to walk a distance of 40 m without walking aids, and (vi) informed consent. Excluded were patients with a history of a walking deficit before the age of 10. Thirty WD-patients were consecutively recruited.

After all, 30 WD-patients had been investigated an age-and sex-matched control group was recruited from the hospital staff or relatives of the authors. Relatives of patients were excluded from the control group because of the possibility to be a gene carrier.

### 2.2. Neurological Examination and Clinical Scores

WD-patients and controls underwent a detailed neurological investigation. Seven motor symptoms (dystonia, dysarthria, bradykinesia, tremor, gait disturbance, oculomotor deficits, ataxia of extremities) and three non-motor symptoms (reflex abnormalities, sensory symptoms, neuropsychological and psychiatric symptoms) were scored whether these symptoms were absent (0) or mildly (1), moderately (2) or severely (3) present. The motor items were summed up to yield a motor score (MotS: 0–21), the three non-motor items were summed up to a non-motor score (N-MotS: 0–9). The sum of MotS and N-MotS yielded the total score (TSC: 0–30). This score has been used in previous studies on WD [14,22,23]. A similar score is used by the Italian study group on liver transplantation in WD [24]. Controls with a score > 0 were excluded from the control group.

Patients were split up into patients who were able to run (RUN-group; n = 25) and patients who were unable to run (NO-RUN-group; n = 5). The RUN-group was split up into 3 further subgroups according to clinical scoring of severity of WD: mildly affected patients (MIL-group; TSC: 0–2; n = 10), moderately affected patients (MOD-group; TSC: 3–6; n = 9), and severely affected patients (SEV-group; TSC > 6; n = 6).

### 2.3. Laboratory Findings

Blood samples were taken for routine monitoring of copper elimination therapy in WD. For the correlation analysis between running parameters and laboratory findings the following parameters were selected: liver enzymes (GOT, GPT, GGT), parameters of copper metabolism (serum level of copper, serum level of ceruloplasmin, copper concentration in the 24 h-urine), Quick’s test, thromboplastin time (PPT), platelet counts and creatinine. The copper content of the 24 h-urine collected under medication was analyzed.

### 2.4. Measurement of Running by Means of the CDG^®^ System

The computer dyno graph (CDG^®^) system (Infotronic^®^; NL-7650 AB Tubbergen, The Netherlands), which consists of a pair of soft tissue shoes with a solid, but flexible plate containing 8 force transducers, allows quantitative measurement of the vertical component of ground reaction forces (GRF-curves) and temporal patterns of foot ground contact. The CDG^®^ shoes are strapped over the street shoes of the patients which had to be tightly fixed to the feet. Thin cables connect the CDG^®^ shoes to a light microprocessor which was tightly attached to a belt strapped around the belly of the patient. Patients and controls were allowed to perform up to 3 short test-runs of less than 8 m to become familiar with the set-up and the safety limits.

Patients and controls were instructed that safety had a higher priority than speed of running. They had to run straight forward for more than 40 m along a broad walkway connecting two buildings. A trial was accepted for further analysis when within the trial a segment of running could be detected with at least 20 cyclic alternations from one leg to the other which were separated by intervals during which both legs did not touch the ground. Thus at least 10 running steps with the right and 10 with the left leg had to be performed. Up to 3 trials were recorded, the first successful segment of running was used for further analysis. For each patient the distance of the selected segment of running was determined and the time and number of steps needed to run along this segment. If the segment was shorter than 40 m, results were extrapolated to a segment of 40 m length for sake of comparison.

The CDG^®^ system analyses a variety of data. From this data pool the following parameters were used: time (in second) needed to run a distance of 40 m (DUR), number of steps needed for running 40 m (NST) to calculate step frequency of running (=number of steps per second = cadence (CAD) = NST/DUR) and speed of running (SPR = 40 m/DUR).

For each selected ground contact (GRF-curve) the time to peak (PT), the peak amplitude (PA), and the duration of ground contact (DFC) were determined (Figure 1). Mean values over all selected steps per side as well as the corresponding standard deviations were calculated for each patient. For each subject and each foot, the GRF-curves of at least 10 ground contacts per side during running were time normalized (=normalized single stance time = NSST) and then superimposed.

For each subject and each foot, the variability of time to peak (PTSD) was calculated as the standard deviation of the PTs of all GRF-curves of the running segment being selected for running analysis.

### 2.5. Statistics

Comparison between data of WD-patients and controls were performed non-parametrically using the Kendell-tau-test. Non-parametric correlations were calculated using Spearman’s rho. An ANOVA was performed to study the influence of severity of symptoms on the ability to run and running parameters. All tests were part of the commercially available statistics package SPSS (version 25: IBM Analytics, Armonk, NY, USA).

## 3. Results

### 3.1. Demographical Data of Patients and Controls

WD-patient and control group were perfectly sex-matched (11 females and 19 males) and did not significantly differ in age, body height and body weight (see Table 1). WD-patients had a mean age at diagnosis of 22 years (SD: 6.8 years). Duration of treatment varied between 31 and 376 months.

### 3.2. Inability to Run of WD-Patients

During running of a normal subject or mildly affected WD-patient the ground is touched alternatingly with the right and the left foot (Figure 2). The time periods of ground contact are separated by time periods during which both legs are in the air (Figure 2; upper part). In five WD-patients no segment could be detected fulfilling our criteria of running (comp. Methods; NO-RUN-group; see Table 2). These patients were females with only one exception, tended to be more severely affected (TSC in Table 2), and had a significantly (*p* < 0.03) higher N-MotS (Table 2). The GRF-curves of these patients (an example is presented on the right sides of Figure 2 and Figure 3) had a double peak and a shape which is typically recorded during walking [15]. Step frequency of running (cadence = NST/DUR) could be determined for all patients and was not significantly reduced to 2.60 s compared to 2.68 s in the controls.

### 3.3. Comparison of Running Parameters in the RUN-Group and the Controls

Speed of running was 2.28 m/s (=40 m/DUR) in the mean in the RUN-group (n = 25) and significantly (*p* < 0.02) lower than in the controls (2.76 m/s). Time needed to run 40 m (DUR) was significantly longer (*p* < 0.02), number of steps needed to run 40 m (NST) was significantly higher (*p* < 0.02), “active” peak amplitudes (PA) of both sides were significantly lower (right leg: *p* < 0.02; left leg: *p* < 0.04), variability of time to peak (PTSD) was significantly higher for both legs (right leg: *p* < 0.01; left leg: *p* < 0.04) and duration of foot ground contact (DFC) was significantly longer for the right leg (*p* < 0.05) than in the controls (see Table 3).

### 3.4. Influence of Severity of WD on Running Parameters

After the RUN-group had been split-up according to severity into the MIL-, MOD-, and SEV-subgroup an ANOVA did not reveal any influence of severity of WD on any running parameter.

### 3.5. Correlations between Clinical Scores and Running Parameters

In the RUN-group (n = 25) motor score (MotS) did not correlate with any running parameter. With most of the running parameters also the total score (TSC) did not correlate. This is demonstrated for the correlation between speed of running (SPR) and TSC in Figure 4. The total score (TSC) was significantly correlated only with PTSD on the left side (r = 0.389, *p* < 0.037). However, when SPR was also calculated for the attempts of running of the NO-RUN-group and SPR was correlated for all patients (n = 30) with TSC a significant correlation was found (r = 0.437; *p* < 0.026).

However, the non-motor score (N-MotS) was significantly correlated with number of steps (NST: r = 0.577, *p* < 0.001), time needed to run 40 m (DUR: r = 0.643, *p* < 0.001), running speed (SPR: r = 0.643; *p* < 0.001), duration of foot contact (DFC: RL: r = 0.569, *p* < 0.001; LL: r = 0.457; *p* < 0.013), time to peak of the left leg (PT: RL: r = 0.221, *p* < 0.186, n.s.; LL: r = 0.407, *p* < 0.028), and variability of time to peak of both legs (PTSD: RL: r = 0.462, *p* < 0.012; LL: r = 0.502, *p* < 0.006).

### 3.6. Correlation between Laboratory Findings and Running Parameters

No significant difference was found between the laboratory findings in the RUN-and the NO-RUN-group. There was a significant correlation between liver enzymes (GPT, GOT) and cadence (GPT: r = −0.397, *p* < 0.033; GOT: r = −0.506, *p* < 0.005; Figure 5). The higher the serum level of creatinine the higher was the variability of time to peak (PTSD: r = 0.530; *p* < 0.003; right leg).

## 4. Discussion

In the present pilot study on quantitative measurement of running in long-term treated WD-patients a mild reduction of speed of running and a high impact of non-motor symptoms on running is demonstrated.

### 4.1. Ability to Run in Normal Subjects and WD-Patients

Running is a complex bipedal motor task which has to be learned after the ability of walking has been acquired [20]. By about 3 years of age, lower limb joint motion during walking and running develop adult-like consistency [25]. Neurological WD does usually not become manifest in early infancy [5]. Thus, there is good reason to assume that all our WD-patients had acquired normal running ability during childhood (comp. inclusion criteria).

At the age of 3, step frequency of running (cadence) is close to 4 Hz [26]. With increasing age, increasing step length, and increasing body weight step frequency of running decreases and reaches a minimum of around 2.5 Hz at the age of 12 to 16 years followed by a mild increase over the next decades up to 2.8 Hz [26]. Mean step frequency of running in the control group was (38.9/14.5 = 2.68 Hz) matching previously reported normal values [26]. Mean step frequency of the RUN-group was only slightly, non-significantly lower (45.5/17.5 = 2.60 Hz).

Only 16% of the WD-patients were unable to run under our test conditions. This is in line with the previously reported observation that the central pattern generator of walking appears to be unimpaired in about 80% of the long-term treated WD-patients [15].

This is an important message for WD-patients when they are diagnosed. At diagnose WD-patients are usually severely sick. The knowledge that there is a reasonable perspective that they can participate in normal life and will be able to perform sports activities again is really helpful to motivate these young patients who heavily rely on the regular intake of sufficiently high medication. This message showed to communicated to the patients by their treating physicians.

### 4.2. Step Frequency of Running and Vertical Stiffness in WD-Patients and Controls

Step frequency of running can be estimated by the formula fs = 12∗π km (where *k* is the vertical stiffness and *m* is the body mass) as long as speed of running is less than 11 km/h [26]. Speed of running in the controls was 40 m/14.5 s = 9.93 km/h and 40 m/17.5 s = 8.23 km/h in the WD-patients. Since bodyweight was nearly the same in patients and controls it can be concluded that the vertical stiffness was lower in the patients than in the normal subjects. This reduction of stiffness goes along with higher variability of times to peak of GRF-curves and their shapes (Figure 3 middle part).

### 4.3. Ground Reaction Force Curves in WD-Patients and Controls

Ground reaction forces (GRF-curves) have gained high interest in engineering. Constructors of robots were interested to understand the transition from walking to running to increase speed of propagation of up-right robots [27,28]. Therefore GRF-curves were studied in detail [29]. During running the center of mass performs regular up and down (and left/right) oscillations which can be simulated by a mass-spring system [MSS]; [29,30]. Such a single body MSS produces sinusoidal GRF-curves with one peak as observed in normal subjects and mildly affected WD-patients (see Figure 2 and Figure 3, left side). However, in robots GRF-curves often have two peaks (see Figure 2 in [29]) as observed in moderately affected WD-patients (see Figure 2 and Figure 3, middle part). These two peaks have been called first and second peak, passive and active peak, impact, and propulsion peak (for an overview see [29]). A simple one-body MSS can only predict the active peak but is not capable of accurately predicting the impact (passive) peak [29,30,31,32]. Two [33,34,35] and three-body models [36] have been developed to simulate the heel strike during running. Meanwhile “the four-body model” developed by Liu and Nigg (LN-model, [37]) is probably the most widely used multi-body MSD model of the human body during hopping and running” [29].

In the moderately affected WD-patients, the impact peak can clearly be distinguished (Figure 3 middle part) indicating the influence of several body segments on the GRF-curves in more affected WD-patients. Controls and mildly affected WD-patients can coordinate and stiffen the feet, lower and upper legs and the trunk to one unit (one mass-spring system) bouncing up and down with high precision as can be seen in Figure 2 and Figure 3 on the left side. Due to reduced stiffness and reduced coordination in the moderately affected patients this precision gets lost, the GRF-curves become more variable (see Figure 2 and Figure 3 middle parts) and the impact peaks more prominent. Therefore, reduced step frequency of running, higher variability of GRF-curves and prominent impact peak reflect a coordination deficit during running in more affected patients with WD.

### 4.4. Running Parameters and Clinical Findings

Running turned out to be highly sensitive to the presence of non-motor symptoms which indicate additional neurological diseases as myelopathies and polyneuropathies. But cognitive impairment may also be caused by side effects of non-Wilson specific medication. Step frequency and time to run a distance of 40 m increased with increasing non-motor score. In contrast, no correlation was found between the motor score and all running parameters. This implies that running is a sensitive test in WD to detect other disabling causes than WD.

### 4.5. Running Parameters and Laboratory Findings

Step frequency (cadence) of running was significantly negatively correlated with serum levels of liver enzymes. However, this does not reflect an affection of the patients by a subclinical hepatoencephalopathy. In most of the patients, the liver enzymes were within normal limits. It rather reflects a tight well-known relationship between liver enzymes and body weight. Increase of body weight reduces step frequency (see Section 2).

## 5. Conclusions

In contrast to the hypothesis mentioned in the introduction, that involvement of the basal ganglia, the cerebellum and the brainstem nuclei will lead to a frequent impairment of running in WD only 16% of the long-term treated WD-patients in the present study were unable to run. This is relevant knowledge for WD-patients and their treating physicians to increase adherence to therapy.

The CDG^®^ system was sensitive enough to demonstrate the negative influence of unspecific non-motor symptoms on running in WD. But no correlation was found between running parameters and the WD-specific motor score. Therefore, running does not seem to be sensitive enough for monitoring WD-therapy.

## 6. Strengths and Shortcomings of the Present Study

The strength of the present study is that it is a controlled pilot study on running in WD. It yields the perspective to WD-patients that under sufficiently high long-term medication most of them will be able to perform sports activities which is a relevant aspect of quality of life.

The CDG^®^ system allows detailed analysis of ground reaction forces, but not the measurement of foot, leg and hip angles and the trajectory of the center of mass during running. Phase plots of displacement angles would probably have shown the coordination problem of different parts of the body more clearly. But compared to a marker-based gait analysis system the CDG^®^ system can be applied much more easily.

## Figures and Tables

**Figure 1 sports-10-00011-f001:**
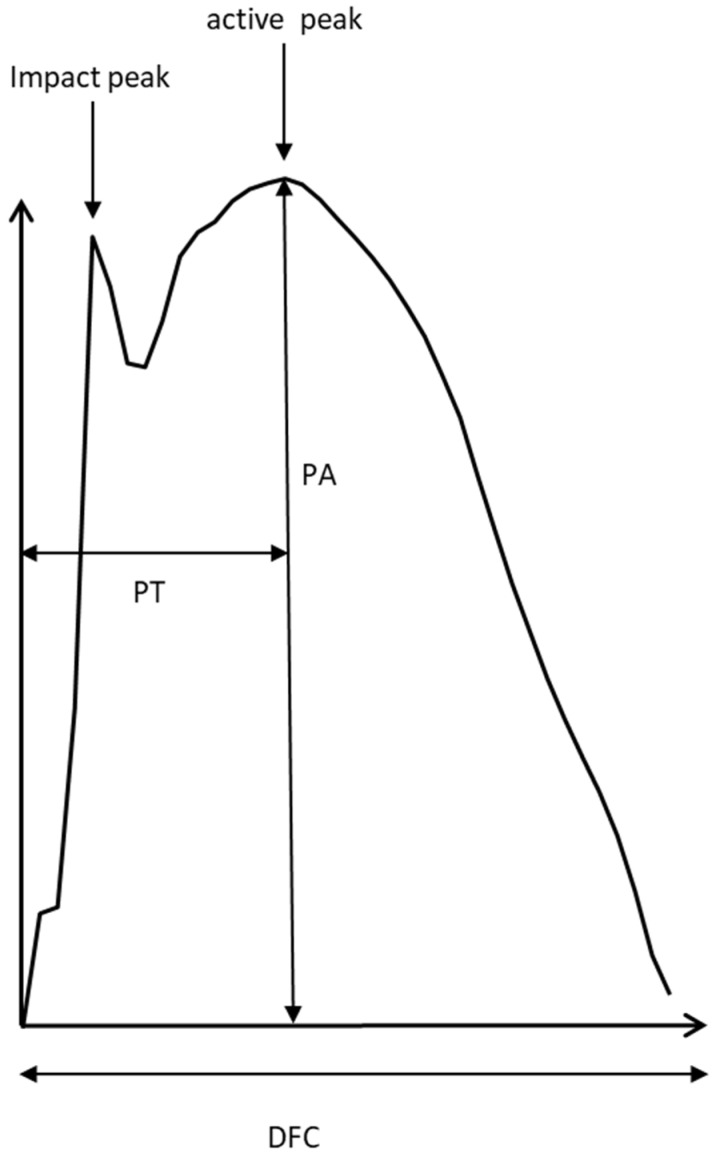
Schematic drawing of a GRF-curve of running: two peaks can be distinguished: the impact and the active peak. The parameters PA (peak amplitude), time to peak (PT) and duration of foot contact (DFC) were determined for each running step.

**Figure 2 sports-10-00011-f002:**
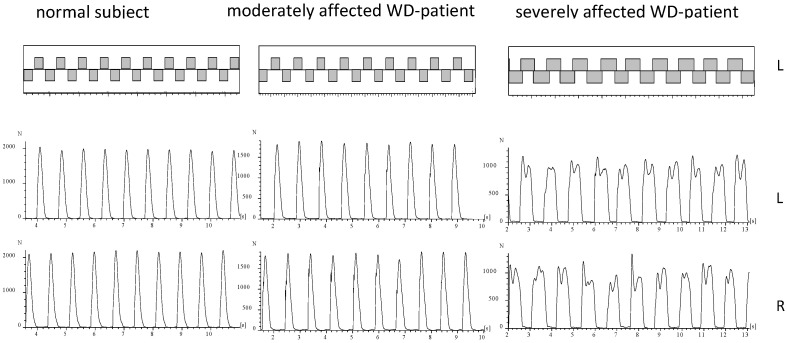
About 10 running steps of the right and the left leg are presented for a normal subject (left side), a moderately affected WD-patient (middle part) and of a severely affected WD-patient (right side). In the upper part the temporal pattern of foot contacts is shown. In the lower part the corresponding GRF-curves are presented. The GRF-curves of the moderately affected patient (middle part) reveals clear impact peaks. The GRF-curves of the severely affected patient (right side) have double peaks as observed during walking.

**Figure 3 sports-10-00011-f003:**
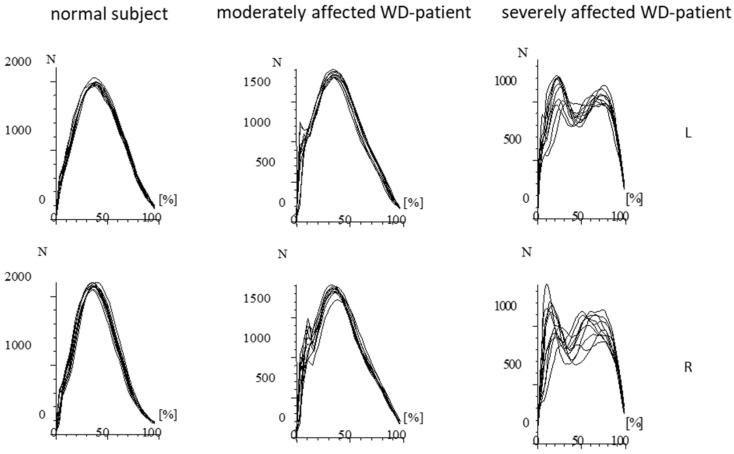
Superposition of the GRF-curves presented in Figure 2 after time normalization. The RF-curves of the normal subject are highly reproducible and performed with little variation. In the moderately affected patient, the variability of the GRF-curves is much higher. Impact peaks can clearly be distinguished. In the severely affected patient, double peaks consistently occur. (N = Newton; L = left foot; R = right foot).

**Figure 4 sports-10-00011-f004:**
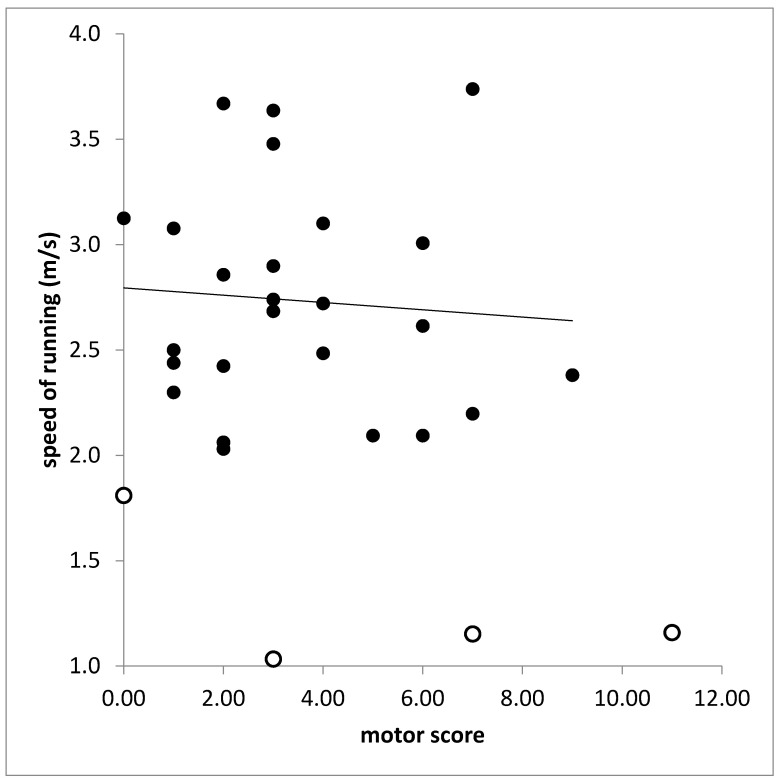
Missing correlation between speed of running (ordinate) and total score (TS). Open symbols indicate patients who belonged to the NO-RUN-group.

**Figure 5 sports-10-00011-f005:**
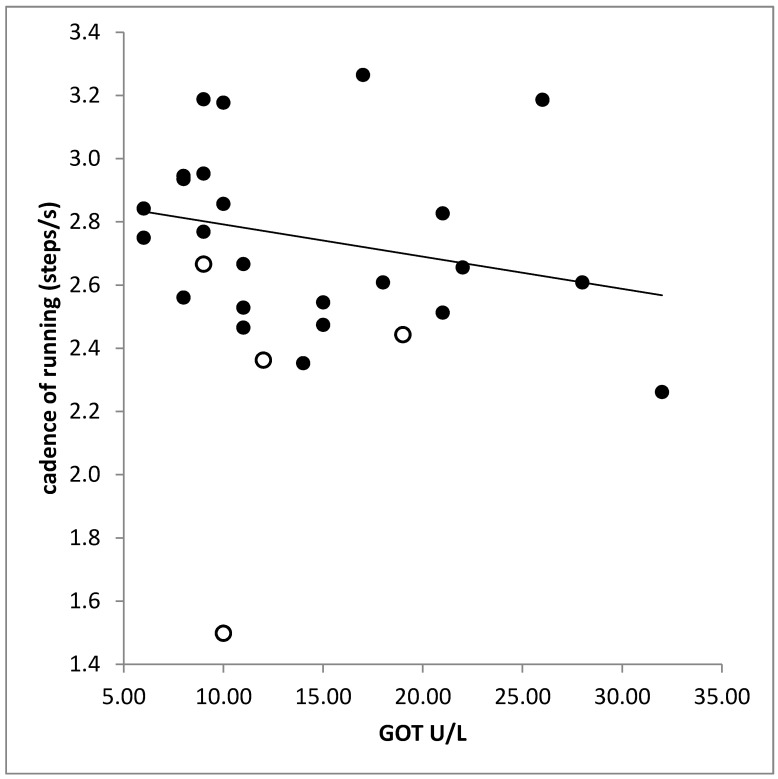
Significant negative correlation between cadence (step frequency of running) and the serum level of the liver enzyme GOT. Open symbols indicate patients who belonged to the NO-RUN-group.

**Table 1 sports-10-00011-t001:** Comparisons of demographical data of WD-patients and controls.

Parameters		WD-Patients n = 30	Control Subjects n = 30	*p* Value
age (years)	mean (SD) range	34.2 (11.0) 15–56	32.8 (8.3) 14–51	*p* = 0.571 n.s.
sex	m/f	19/11	19/11	*p* = 1.0 n.s.
body height (cm)	mean (SD) range	177.8 (10.3) 156–196	176.9 (9.5) 160–194	*p* = 0.73 n.s.
body weight (kg)	mean (SD) range	72.7 (1.5) 48–126	73.6 (14.8) 51–105	*p* = 0.83 n.s.

mean = mean value; SD = standard deviation; n.s. = not significant.

**Table 2 sports-10-00011-t002:** Comparisons of clinical data of the RUN- and NO-RUN-group.

Parameters		RUN Group	NO-RUN-Group	*p* Value
sex distribution	m/f	18/7	1/4	*p* = 0.054; n.s.
TSC	mean (SD)	4.2 (3.2)	7.0 (6.2)	*p* = 0.14; n.s.
MotS	mean (SD)	3.5 (2.4)	4.8 (4.2)	*p* = 0.34; n.s.
N-MotS	mean (SD)	0.7 (1.2)	2.2 (1.9)	*p* < 0.03

TSC = total score; MotS = motor score; N-MotS = non-motor score; mean = mean value; SD = standard deviation.

**Table 3 sports-10-00011-t003:** Comparisons of running data of WD-patients and controls.

Parameters		RUN-Group	Control Subjects	*p* Value
DUR (in second)	mean (SD)	17.5 (7.0)	14.5 (1.7)	*p* < 0.02
NST	mean (SD)	45.5 (14.2)	38.9 (4.9)	*p* < 0.02
DFC (in second)	mean (SD)	RL: 0.36 (0.12) LL: 0.35 (0.12)	RL: 0.31 (0.03) LL: 0.31 (0.03)	*p* < 0.05 *p* = 0.06; n.s.
PA (in N)	mean (SD)	RL: 1579 (418) LL: 1553 (457)	RL: 1843 (429) LL: 1785 (399)	*p* < 0.02 *p* < 0.04
PTSD (in ms)	mean (SD)	RL: 16.1 (14.2) LL: 16.8 (19.5)	RL: 9.2 (2.5) LL: 9.1 (2.6)	*p* < 0.01 *p* < 0.04

mean = mean value; SD = standard deviation; DUR = time in seconds to run 40 m; NST = number of steps to run 40 m; DFC = duration of foot ground contact in seconds; PA = peak amplitude in N; PTSD = variability of time to peak in ms (see Methods); RL = right leg; LL = left leg).

## Data Availability

Data available on request due to restrictions e.g., privacy or ethical. The data presented in this study are available on request from the corresponding author.

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
