# Peer review of "Analysis of Running in Wilson’s Disease"

_sports, 2022, doi:10.3390/sports10010011_

Round 1

Reviewer 1 Report

Paper entitled: ”Analysis of running in Wilson´s disease” (sports-1501321) presents study about the analysis the ability of long-term treated patients with Wilson´s disease to run a distance of 40 meters. The work is well written, but please explain more clearly what is the purpose of conducting this research and comparing it between the group of patients assigned to the experiment. The research is pilot, but the scientific significance of the research is not fully understood, in addition to excluding the running test for monitoring WD-therapy and long-term outcomes. However, there are some issues which should be addressed before publication in Sports.

L98: please extend the abbreviation TSC here

L184: please remove the double dot

L286 and 279 twice repeated sentence that there were no sponsors

Author Response

Paper entitled: ”Analysis of running in Wilson´s disease” (sports-1501321) presents study about the analysis the ability of long-term treated patients with Wilson´s disease to run a distance of 40 meters. The work is well written, but please explain more clearly what is the purpose of conducting this research and comparing it between the group of patients assigned to the experiment. The research is pilot, but the scientific significance of the research is not fully understood, in addition to excluding the running test for monitoring WD-therapy and long-term outcomes. However, there are some issues which should be addressed before publication in Sports.

L98: please extend the abbreviation TSC here

L184: please remove the double dot

L286 and 279 twice repeated sentence that there were no sponsors

Reviewer 1 is absolutely right that one of the main reasons to perform the study has not been emphasized clearly:

It is to enhance the compliance of the patients.

The perspective at onset of therapy for these young severely affected patients with WD that a fairly normal performance sports activities will become possible with good adherence to therapy is highly relevant for patients and treating physicians. 

TSC had been explained in line 92.

Now we also extend the abbriviations in line 98 and 99.

This is corrected.

This sentence is now mentioned only once.

Reviewer 2 Report

COMMENT:

 Sara Samadzadeh et al. demonstrated quantitative measurement of running in long-term treated WD-patients, a mild reduction of speed of running and a high impact of non-motor symptoms on running. They concluded that running is not a useful test for monitoring therapy and outcome in WD. Although no significant difference was found between WD-patients and controls, this kind of clinical studies on running of neurological patients are rare. So, I recommend accepting this manuscript as a regular type of manuscript after the minor revisions noted below.

Minor comments.

  1. Page4 Fig1, Page5 Fig2, and Page6 Fig3, please put label to vertical and horizontal axes like shown as Figs 4 and 5.    For example, "Time (min)" in horizontal axes of Fig 2.
  1. Please check and unify the “Abbreviations”. For example, “variability of time to peak (PTSD)” on page 6 line 166, “PTSD=standard deviation of time to peak in ms” on page 10.

Author Response

Sara Samadzadeh et al. demonstrated quantitative measurement of running in long-term treated WD-patients, a mild reduction of speed of running and a high impact of non-motor symptoms on running. They concluded that running is not a useful test for monitoring therapy and outcome in WD. Although no significant difference was found between WD-patients and controls, this kind of clinical studies on running of neurological patients are rare. So, I recommend accepting this manuscript as a regular type of manuscript after the minor revisions noted below.

Minor comments.

  1. Page4 Fig1, Page5 Fig2, and Page6 Fig3, please put label to vertical and horizontal axes like shown as Figs 4 and 5.    For example, "Time (min)" in horizontal axes of Fig 2.
  1. Please check and unify the “Abbreviations”. For example, “variability of time to peak (PTSD)” on page 6 line 166, “PTSD=standard deviation of time to peak in ms” on page 10.

As shown in Table 3 several parameters of  running were significantly different between WD-patients and controls.

As suggested by Reviewer 2 we have put labels to the vertical and horizontal axes of Fig. 1,2,3.

We checked the abbreviations once again.

Reviewer 3 Report

The authors haven't analyzed adequately the reason of the negative correlation between the running parameters and WD-specific motor scores

Author Response

The authors haven't analyzed adequately the reason of the negative correlation between the running parameters and WD-specific motor scores

Reviewer 3 is absolutely right:

As mentioned in line 175 and 176 we did not find a correlation between the motor score (MotS) or the total score (TSC) and running parameters. This correlation analysis was restricted to the RUN-group. That had not been mentioned clearly.

However, when the correlation was performed for all WD-patients a significant negative correlation was found between speed of running and for the total score. That result had been omitted in the old manuscript and is now added.  

We are thankful for this helpful comment.